# The effect of liver enzymes on body composition: A Mendelian randomization study

Junxi Liu[1], Shiu Lun Au Yeung[1], Man Ki Kwok[1], June Yue Yan Leung[1], Lai Ling Hui[1,2], Gabriel Matthew Leung[1], C. Mary Schooling[1,3]*

1 School of Public Health, Li Ka Shing Faculty of Medicine, The University of Hong Kong, Hong Kong SAR, China, 2 Department of Paediatrics, Faculty of Medicine, the Chinese University of Hong Kong, Hong Kong SAR, China, 3 City University of New York Graduate School of Public Health and Health Policy, New York, New York, United States of America

* cms1@hku.hk

## Abstract

### Background

Higher alanine transaminase (ALT), indicating poor liver function, is positively associated with diabetes but inversely associated with body mass index (BMI) in Mendelian randomization (MR) studies, suggesting liver function affects muscle mass. To clarify, we assessed the associations of liver enzymes with muscle and fat mass observationally with two-sample MR as a validation.

### Methods

In the population-representative "Children of 1997" birth cohort (n = 3,455), we used multivariable linear regression to assess the adjusted associations of ALT and alkaline phosphatase (ALP) at ~17.5 years with muscle mass and body fat percentage observationally. Genetic variants predicting ALT, ALP and gamma glutamyltransferase (GGT) were applied to fat-free and fat mass in the UK Biobank (n = ~331,000) to obtain unconfounded MR estimates.

### Results

Observationally, ALT was positively associated with muscle mass (0.11 kg per IU/L, 95% confidence interval (CI) 0.10 to 0.12) and fat percentage (0.15% per IU/L, 95% CI 0.13 to 0.17). ALP was inversely associated with muscle mass (-0.03 kg per IU/L, 95% CI -0.04 to -0.02) and fat percentage (-0.02% per IU/L, 95% CI -0.03 to -0.01). Using MR, ALT was inversely associated with fat-free mass (-0.41 kg per 100% in concentration, 95% CI -0.64 to -0.19) and fat mass (-0.58 kg per 100% in concentration, 95% CI -0.85 to -0.30). ALP and GGT were unclearly associated with fat-free mass or fat mass.

**Data Availability Statement:** The data that support the findings of this study are available on request from the "Children of 1997" data access committee: aprmay97@hku.hk. The data are not publicly available due to the participants could be

identifiable from such extensive data which would comprise participant privacy. The datasets analyzed during the current MR study are publicly available summary data. These datasets were derived from the public domain resources: a publicly available GWAS study https://www.nature.com/articles/ng.970 and the UK Biobank GWAS; http://www.nealelab.is/blog/2017/9/11/details-and-considerations-of-the-uk-biobank-gwas.

**Funding:** This work is a substudy of the "Children of 1997" birth cohort which was initially supported by the Health Care and Promotion Fund, Health and Welfare Bureau, Government of the Hong Kong SAR [HCPF grant 216106] and reestablished in 2005 with support from the Health and Health Services Research Fund, Government of the Hong Kong SAR, [HHSRF grant 03040771]; the Research Fund for the Control of Infectious Diseases in Hong Kong, the Government of Hong Kong SAR [RFCID grant 04050172]; the University Research Committee Strategic Research Theme (SRT) of Public Health, the University of Hong Kong. The Biobank Clinical follow-up was partly supported by the WYNG Foundation. The funders had no role in study design, data collection and analysis, decision to publish, or preparation of the manuscript.

**Competing interests:** The authors have declared that no competing interests exist.

## Conclusion

ALT reducing fat-free mass provides a possible pathway for the positive association of ALT with diabetes and suggests a potential target of intervention.

## Introduction

Observationally, poorer liver function, particularly nonalcoholic fatty liver disease, is associated with a higher risk of type 2 diabetes mellitus (T2DM).[1] Mendelian randomization (MR) studies, taking advantage of the random allocation of genetic endowment at conception to obtain un-confounded estimates, [2] have clarified the role of liver function in T2DM. Specifically, these studies suggest that higher alanine aminotransferase (ALT) [3, 4] or aspartate aminotransferase (AST) [4] rather than other measures of liver function, such as glutamyltransferase (GGT), [3, 4] could play a role in T2DM. However, modifiable targets on the pathway from poor liver function to T2DM are unclear and worthy of exploration. Recently, MR studies using different data sources have found ALT inversely associated with body mass index (BMI), indicating higher ALT might reduce BMI.[5, 6] This finding appears to contradict observational studies that show adiposity associated with poor liver function.[7] However BMI does not distinguish muscle mass from fat mass.[8] Nevertheless, ALT reducing the muscle mass component of BMI would be consistent with ALT increasing the risk of diabetes, given low muscle mass is a potential cause of diabetes.[9] Observationally, liver function is associated with muscle mass, although these studies are not always consistent.[10, 11] These inconsistencies could be due to confounding by lifestyle, health status, and socioeconomic position (SEP), or to selection bias in studies conducted in patients.

To clarify the roles of liver enzymes, indicating liver function, in body composition in the absence of experimental evidence, we conducted two analyses with different assumptions and study designs. Observationally, we examined the associations of ALT and alkaline phosphatase (ALP) with commonly used measures of muscle mass, i.e., muscle mass and grip strength, and fat percentage in a young population in a setting with little socioeconomic patterning of obesity, so as to reduce confounding by poor health and SEP, i.e., in Hong Kong's "Children of 1997" birth cohort.[12] Given the differences in body composition by sex, we also examined whether the associations differed by sex because such differences are likely interpretable even when other associations are confounded.[13] To validate the impact of liver enzymes on body composition, we also used an MR design to assess the effects of genetically predicted ALT, ALP, and GGT [14] on body composition (fat-free mass, grip strength, and fat mass) from the UK Biobank.[15] Differences by sex were investigated given sex disparities in liver disease [16, 17] and body composition [18] have been observed previously.

## Material and methods

### Ethics statement

Ethical approval for the study, including comprehensive health related analyses, was obtained from Institutional Review Board of the University of Hong Kong/Hospital Authority Hong Kong West Cluster (HKU/HA HKW IRB). Informed written consent was obtained from the parents/guardians, or from the participant if 18 years or older, before participation in the Biobank Clinical Follow-up.

The MR study only uses published or publicly-available data. No original data were collected for the MR study. Ethical approval for each of the studies included in the investigation can be found in the original publications (including informed consent from each participant).

## Observational study—The "Children of 1997" birth cohort

The "Children of 1997" birth cohort is a population-representative Chinese birth cohort (n = 8,327) which included 88% of all births in Hong Kong from 1 April 1997 to 31 May 1997. [19] The study was initially established to examine the effects of second-hand smoke exposure and breastfeeding on health services utilization to 18 months. Participants were recruited at the first postnatal visit to any of the 49 Maternal and Child Health Centers in Hong Kong, which parents of all newborns are strongly encouraged to attend to obtain free preventive care and vaccinations for their child/children up to 5 years of age. Information including parental characteristics (maternal age, paternal age, parental smoking, and parental education) and infant characteristics (birth weight, gestational age, and sex) was obtained from a self-administered questionnaire in Chinese at recruitment and subsequent routine visits. Parental occupation, type of housing and income were also recorded.

At the Biobank Clinical follow-up at age ~17·5 years, as a compromise between cost and comprehensiveness, liver enzymes were assessed from ALT and ALP, a marker of hepatocyte integrity and a marker of cholestasis.[20] These were analyzed using the Roche Cobas C8000 System, a discrete photometric chemistry analyzer, with International Federation of Clinical Chemistry standardized method with pyridoxal phosphate and substrates of L-alanine and 2-oxoglutarate for ALT, and an optimized substrate concentration and 2-amino-2-methyl-1-propanol as buffer plus the cations magnesium and zinc for ALP. These analyses were conducted at an accredited laboratory serving a teaching hospital in Hong Kong. Body composition indices including muscle mass and fat percentage were measured using bioimpedance analysis by a Tanita segmental body composition monitor (Tanita BC-545, Tanita Co., Tokyo, Japan). Grip strength was measured by the Takei T.K.K.5401 GRIP D handgrip dynamometer (Takei Scientific Instruments Co. Ltd, Tokyo, Japan). All anthropometric measurements were made by trained research assistants following specific standard protocols.

**Exposures—Liver enzymes.**   Liver function at ~17·5 years was assessed from plasma ALT (IU/L) and ALP (IU/L).

**Outcomes—Body composition.**   Muscle was assessed from muscle mass (kg) and dominant hand grip strength (kg). Fat mass was assessed from body fat percentage.

## Mendelian randomization study

**Exposure—Genetic predictors of liver enzymes.**   Single nucleotide polymorphisms (SNPs) predicting plasma log-transformed ALT, ALP and GGT at genome-wide significance (p-value$<5\times10^{-8}$) adjusted for age and sex were obtained from the largest available genome-wide association study (GWAS) of plasma levels of liver enzymes comprising 61,089 adults (~86% European, mean age 52.8 years, 50.6% women). The estimate for each SNP obtained from the GWAS represents the % change in concentration of liver enzyme in plasma (effect size) per copy of the effect allele.[14] For SNPs in linkage disequilibrium ($R^2>0.01$), we retained SNPs with the lowest p-value using the "*Clumping*" function of MR-Base (*TwoSampleMR*) R package, based on the 1000 Genomes catalog.[21] Whether any of the selected SNPs were associated with potential confounders was assessed from their Bonferroni corrected associations with height, alcohol use (intake frequency and intake versus 10 years previously), smoking (current smoking and past smoking), education, financial situation, physical activity (moderate and vigorous physical activity), and age of puberty (menarche and voice breaking) in the UK

Biobank. (ALT, 10 traits × 4 SNPs, p-value<1×10$^{-3}$; ALP, 10 traits × 14 SNPs, p-value<3×10$^{-4}$; GGT, 10 traits × 26 SNPs, p-value<1×10$^{-4}$). Additionally, we assessed the pleiotropic effects (related to body compositions directly rather than through liver enzymes) of the selected SNPs from comprehensive curated genotype to phenotype cross-references, i.e., Ensembl (http://www.ensembl.org/index.html) and the GWAS Catalog (https://www.ebi.ac.uk/gwas/). Lastly, we considered SNPs in the *ABO* and *GCKR* genes as potentially pleiotropic SNPs because these genes have many different effects that could possibly affect body composition directly rather than via liver enzymes.

**Outcome—Genetic associations with body composition.**   Genetic associations with fat-free mass (kg), grip strength (kg) (left and right hand), and fat mass (kg) were obtained from UK Biobank (~331,000 people of genetically verified white British ancestry) where the associations were obtained from multivariable linear regression adjusted for the first 20 principal components, sex, age, age-squared, the sex and age interaction and the sex and age-squared interaction.[15]

## Statistical analyses

**Observational analyses.**   In the "Children of 1997" birth cohort, baseline characteristics were compared between cohort participants who were included and excluded using chi-squared tests and Cohen effect sizes which indicate the magnitude of differences between groups independent of sample size. Cohen effect sizes are usually categorized as 0.20 for small, 0.50 for medium and 0.80 for large, but when considering categorical variables they are categorized as 0.10 for small, 0.30 for medium and 0.50 for large.[22] The associations of body composition with potential confounders were assessed using independent t-tests or analysis of variance. We assessed the associations of liver enzymes with body composition indices using multivariable linear regression, adjusted for household income, highest parental education, type of housing, highest parental occupation, second-hand and maternal smoking, height and sex.Bonferroni corrected p-value (<0.003, 2 exposures, 3 outcomes, and sex-specific estimates (n = 3)) was additionally used to account for multiple testing. For a small proportion of the observations, ALT was lower than 10 IU/L (n = 254, 7.3%), without specific value, were fixed at 5 IU/L. We also assessed whether associations differed by sex from the significance of interactions adjusted for the other potential confounding interactions by sex.

**Mendelian randomization analyses.**   We assessed the strength of the genetic instruments based on the *F*-statistic, where a higher *F*-statistic indicates a lower risk of weak instrument bias.[23] All SNPs were aligned according to the effect allele frequency for both the exposure and outcome.

We obtained the effects of liver enzymes on body composition indices based on meta-analysis of SNP-specific Wald estimates (SNP-outcome association divided by SNP-exposure association) using inverse variance weighting (IVW) with multiplicative random effects for 4+ SNPs, where the variance of each SNP-specific Wald estimate is multiplied by Cochran's Q/(number of instruments minus 1) when larger than 1 to allow for heterogeneity assuming balanced pleiotropy and zero average pleiotropic effect of variants.Fixed effects estimateds were used for 3 SNPs or fewer. As such, both the fixed- and the multiplicative random- effects models give the same point estimate, [24] but usually different confidence intervals. Heterogeneity was assessed using the $I^2$ statistic where a high $I^2$ may indicate the presence of invalid SNPs.[25] Power calculations were performed using the approximation that the sample size for Mendelian randomization equates to that of the same regression analysis with the sample size divided by the r$^2$ from regression of genetic variant on exposure.[26] Differences by sex were also assessed.

**Sensitivity analyses.**   First, we repeated the analyses excluding potentially pleiotropic SNPs and those associated with confounders in the UK Biobank. Second, we used a weighted median (WM) which may generate correct estimates as long as >50% of the weight is contributed by valid SNPs.[27] Third, we used MR-Egger which generates correct estimates even when all the SNPs are invalid instruments as long as the instrument strength independent of direct effect (InSIDE) assumption, that the pleiotropic effects of genetic variants are independent of the instrument strength, is satisfied.[25] A non-null intercept from MR-Egger indicates potentially directional pleiotropy and an invalid IVW estimate.[27] Finally, as an additional check on the validity of the MR estimates, we used Mendelian Randomization Pleiotropy RESidual Sum and Outlier (MR-PRESSO), which precisely detects and corrects for pleiotropic outliers assuming >50% of the instruments are valid, balanced pleiotropy and the InSIDE assumption are satisfied. Ideally, it gives a causal estimate with less bias and better precision than IVW and MR-Egger additionally assuming ≤10% of horizontal pleiotropic variants.[28]

All statistical analyses were conducted using R version 3·4·2 (R Foundation for Statistical Computing, Vienna, Austria). The R packages *MendelianRandomization*[29] and *MRPRESSO* [28] were used to generate the estimates.

# Results

## Children of 1997

Of 8,327 initially recruited, 6,850 are contactable and living in Hong Kong, of whom 3,460 (51%) took part in the Biobank Clinical follow-up. Of these 3,460, 3,455 had measures of muscle mass, grip strength or fat percentage, as shown in Fig 1. [30] The mean and standard deviation (SD) of muscle mass, grip strength, and fat percentage were 42.6kg (SD 8.8kg), 25.8kg (SD 8.3kg), and 21.7% (SD 8.8%). Boys had higher muscle mass and grip strength but lower fat percentage than girls, but body composition had little association with SEP (Table 1). There were some differences between participants included and excluded from the study, such as sex, second-hand and maternal smoking exposure, and SEP, but the magnitude of these differences was small (Cohen effect size <0.15) (S1 Table).

Overall, ALT was positively associated with muscle mass (0.11, 95% CI 0.10 to 0.12) and fat percentage (0.15, 95% CI 0.13 to 0.17), whereas, ALP was negatively associated with muscle mass (-0.03, 95% CI- 0.04 to -0.02) and fat percentage (-0.02, 95% CI -0.03 to -0.01). These associations were robust to using a Bonferroni correction (p-value of 0.003). The associations of liver enzymes with muscle mass and fat percentage differed by sex (Table 2). ALT was more strongly positively associated with muscle mass (0.13, 95% CI 0.11 to 0.14) and fat percentage (0.16, 95% CI 0.14 to 0.18) in boys. ALT was not clearly associated with grip strength. ALP was inversely associated with muscle mass (-0.04, 95% CI -0.05 to -0.03), fat percentage (-0.03, 95% CI -0.04 to -0.02), and grip strength (-0.02, 95% CI -0.03 to -0.01) in boys, whereas, ALP was unclearly associated with muscle mass but positively associated with fat percentage (0.030, 95% CI 0.004 to 0.048) and grip strength (0.020, 95% CI 0.001 to 0.033) in girls.

## Mendelian randomization

**Genetic instruments for liver enzymes.**   Altogether, 4 SNPs independently predicting ALT, 14 SNPs independently predicting ALP and 26 SNPs independently predicting GGT at genome-wide significance were obtained (S2 Table).[14] Palindromic SNPs were all aligned according to effect allele frequency (S3 Table). The *F* statistic and variance explained ($r^2$) were 15 and 0.001 for ALT, 158 and 0.035 for ALP, and 45 and 0.019 for GGT, respectively. As such the MR study had 80% power with 5% alpha to detect a difference of 0.15, 0.03 and 0.04 in fat-free mass and fat mass effect size for ALT, ALP, and GGT respectively.

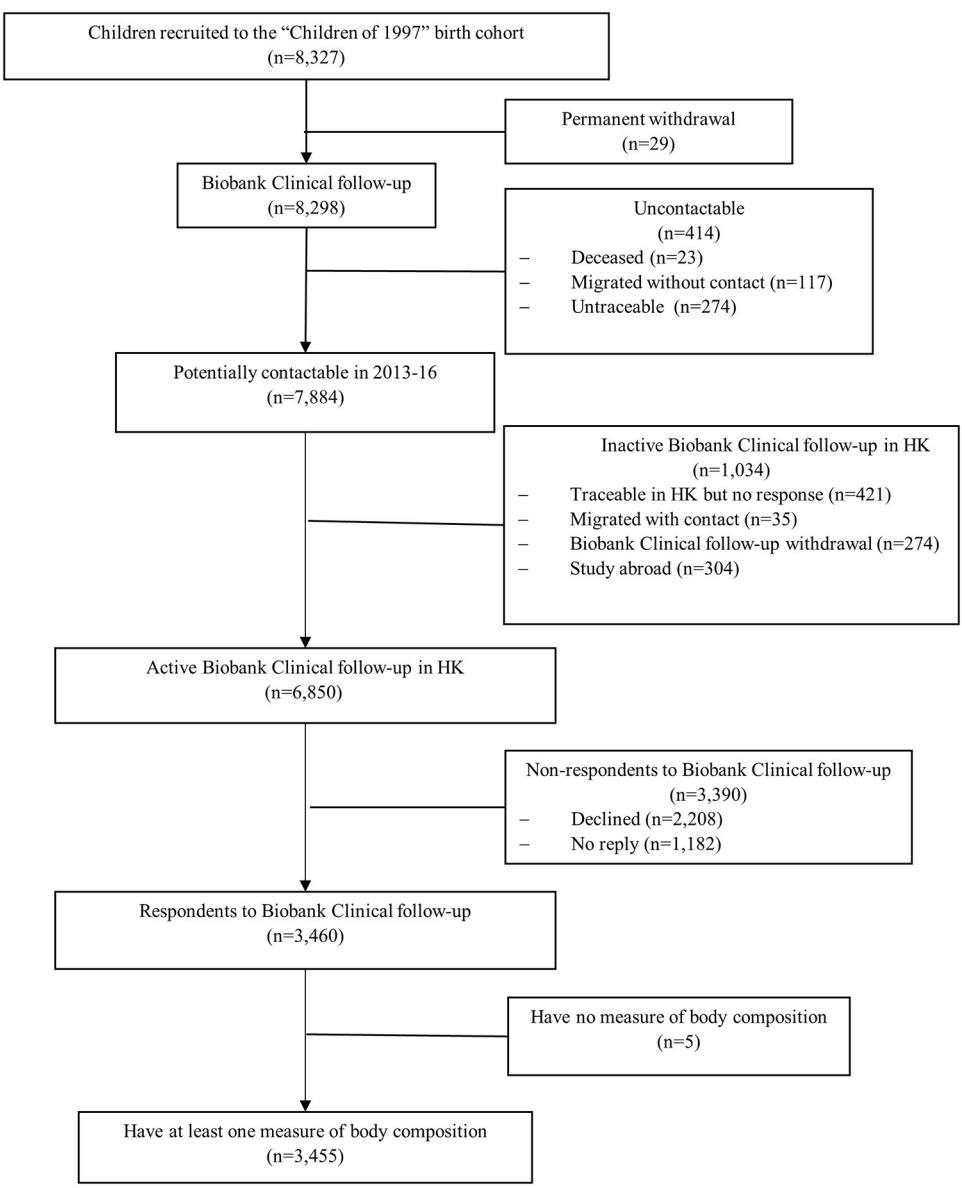

**Fig 1. Flow chart of the Hong Kong's "Children of 1997" birth cohort, Hong Kong, China, 1997 to 2016.**

One SNP, rs2954021 (*TRIB1*), predicting ALT was associated with potential confounders. Seven SNPs, rs174601 (*C11orf10*, *FADS1*, *FADS2*), rs2236653 (*ST3GAL4*), rs281377 (*FUT2*), rs2954021 (*TRIB1*), rs579459 (*ABO*), rs6984305 (*PPP1R3B*) and rs7923609 (*JMJD1C*, *NRBF2*) predicting ALP were associated with potential confounders. Eight SNPs, rs10908458 (*DPM3*, *EFNA1*, *PKLR*), rs12145922 (*CCBL2*, *PKN2*), rs1260326 (*GCKR*), rs1497406 (*RSG1*, *EPHA2*), rs17145750 (*MLXIPL*), rs516246 (*FUT2*), rs7310409 (*HNF1A*, *C12orf27*) and rs754466 (*DLG5*), predicting GGT were associated with potential confounders in UK Biobank at Bonferroni corrected significance (S2 Table).

Among the 4 SNPs predicting ALT, rs2954021 (*TRIB1*) predicts both ALT and ALP, and rs738409 (*PNPLA3*) is highly associated with non-alcoholic fatty liver disease. Among the 14 SNPs predicting ALP, rs281377 (*FUT2*) is highly associated with resting metabolic rate,

**Table 1. Baseline characteristics muscle mass, grip strength, and fat percentage among participants in Hong Kong's "Children of 1997" birth cohort, Hong Kong, China, 1997 to 2016.**

| Characteristics | Muscle mass (kg) | | | | Grip strength (kg) | | | | Fat percentage | | | |
|---|---|---|---|---|---|---|---|---|---|---|---|---|
| | No. | % | Mean (SD) | *P-value*[a] | No. | % | Mean (SD) | *P-value*[a] | No. | % | Mean (SD) | *P-value*[a] |
| Muscle mass (kg) | 3440 | | 42.6 (8.8) | | | | | | | | | |
| Grip strength (kg) | | | | | 3444 | | 25.8 (8.3) | | | | | |
| Fat percentage (%) | | | | | | | | | 3452 | | 21.7 (8.8) | |
| Sex | 3440 | | | <0.001 | 3444 | | | <0.001 | 3452 | | | <0.001 |
| Girl | 1707 | 49.6% | 35.3 (3.4) | | 1710 | 49.7% | 19.9 (4.5) | | 1714 | 49.7% | 28.1 (5.9) | |
| Boy | 1733 | 50.4% | 49.7 (6.3) | | 1734 | 50.3% | 31.6 (7.0) | | 1738 | 50.3% | 15.3 (6.4) | |
| Unknown | 0 | 0.0% | - | | 0 | 0.0% | - | | 0 | 0.0% | - | |
| Second-hand and maternal smoking exposure | 3440 | | | 0.07 | 3444 | | | 0.77 | 3452 | | | 0.17 |
| None | 940 | 27.3% | 42.1 (8.4) | | 939 | 27.3% | 25.6 (8.1) | | 943 | 27.3% | 21.2 (8.5) | |
| Prenatal second-hand smoking | 1275 | 37.1% | 42.7 (8.8) | | 1276 | 37.0% | 26.0 (8.4) | | 1276 | 37.0% | 21.6 (9.0) | |
| Postnatal second-hand smoking | 953 | 27.7% | 43.0 (9.2) | | 956 | 27.8% | 25.7 (8.3) | | 960 | 27.8% | 22.0 (9.0) | |
| Maternal smoking | 128 | 3.7% | 42.7 (8.8) | | 128 | 3.7% | 26.0 (8.2) | | 128 | 3.7% | 22.9 (8.6) | |
| Unknown | 144 | 4.2% | 41.1 (8.6) | | 145 | 4.2% | 25.3 (8.7) | | 145 | 4.2% | 21.9 (9.0) | |
| Highest parental education level | 3440 | | | 0.06 | 3444 | | | 0.12 | 3452 | | | 0.04 |
| Grade< = 9 | 984 | 28.6% | 42.2 (9.1) | | 988 | 28.7% | 25.4 (8.3) | | 989 | 28.7% | 22.2 (9.0) | |
| Grades 10–11 | 1481 | 43.1% | 42.4 (8.6) | | 1483 | 43.1% | 25.7 (8.4) | | 1488 | 43.1% | 21.6 (8.8) | |
| Grades> = 12 | 959 | 27.9% | 43.1 (8.9) | | 957 | 27.8% | 26.3 (8.1) | | 959 | 27.8% | 21.1 (8.7) | |
| Unknown | 16 | 0.5% | 39.7 (7.3) | | 16 | 0.5% | 24.4 (6.8) | | 16 | 0.5% | 23.9 (8.6) | |
| Highest parental occupation | 3440 | | | 0.32 | 3444 | | | 0.04 | 3452 | | | 0.12 |
| I (unskilled) | 98 | 2.8% | 41.9 (9.3) | | 99 | 2.9% | 25.4 (8.6) | | 99 | 2.9% | 21.8 (8.1) | |
| II(semiskilled) | 281 | 8.2% | 43.0 (9.0) | | 283 | 8.2% | 26.4 (8.3) | | 285 | 8.3% | 21.9 (8.8) | |
| III (semiskilled) | 503 | 14.6% | 42.3 (9.0) | | 504 | 14.6% | 25.1 (8.4) | | 503 | 14.6% | 21.5 (8.8) | |
| III (nonmanual skilled) | 876 | 25.5% | 42.4 (8.7) | | 878 | 25.5% | 25.4 (8.1) | | 879 | 25.5% | 22.2 (9.2) | |
| IV (managerial) | 438 | 12.7% | 43.2 (9.5) | | 438 | 12.7% | 26.5 (8.5) | | 439 | 12.7% | 22.2 (8.6) | |
| V (professional) | 794 | 23.1% | 42.8 (8.5) | | 792 | 23.0% | 26.2 (8.2) | | 795 | 23.0% | 21.0 (8.5) | |
| Unknown | 450 | 13.1% | 42.0 (8.5) | | 450 | 13.1% | 25.3 (8.4) | | 452 | 13.1% | 21.5 (9.2) | |
| Household income per head at recruitment | 3440 | | | 0.07 | 3444 | | | 0.16 | 3452 | | | 0.15 |
| First quintile | 566 | 16.5% | 42.0 (8.5) | | 572 | 16.6% | 25.6 (8.5) | | 571 | 16.5% | 21.7 (8.9) | |
| Second quintile | 613 | 17.8% | 41.9 (9.3) | | 613 | 17.8% | 25.0 (8.3) | | 616 | 17.8% | 22.2 (8.7) | |
| Third quintile | 616 | 17.9% | 43.3 (8.8) | | 617 | 17.9% | 26.1 (8.3) | | 618 | 17.9% | 21.8 (9.1) | |
| Fourth quintile | 630 | 18.3% | 42.7 (8.9) | | 629 | 18.3% | 25.9 (8.5) | | 630 | 18.3% | 21.2 (8.7) | |
| Fifth quintile | 644 | 18.7% | 42.9 (8.6) | | 642 | 18.6% | 26.1 (7.9) | | 645 | 18.7% | 21.1 (8.5) | |
| Unknown | 371 | 10.8% | 42.6 (9.0) | | 371 | 10.8% | 26.1 (8.3) | | 372 | 10.8% | 22.2 (9.2) | |
| Type of housing at recruitment | 3440 | | | 0.45 | 3444 | | | 0.44 | 3452 | | | 0.36 |
| Public | 1435 | 41.7% | 42.5 (8.9) | | 1440 | 41.8% | 25.8 (8.5) | | 1445 | 41.9% | 21.9 (9.1) | |
| Subsidized home ownership scheme | 545 | 15.8% | 42.2 (8.8) | | 541 | 15.7% | 25.2 (8.2) | | 544 | 15.8% | 22.0 (8.9) | |
| Private | 1355 | 39.4% | 42.8 (8.8) | | 1358 | 39.4% | 25.9 (8.1) | | 1358 | 39.3% | 21.3 (8.5) | |
| Unknown | 105 | 3.1% | 41.8 (8.8) | | 105 | 3.0% | 25.8 (8.7) | | 105 | 3.0% | 21.2 (8.7) | |

[a] P-values for the associations of body composition with potential confounders were from independent t-tests or from analysis of variance (i.e., smoking, education, occupation, income, and housing).

rs579459 is located in the *ABO* gene whose impact is extensive but unclear. Among the 26 SNPs predicting GGT, rs12968116 (*ATP8B1*) is associated with body height, rs1260326 (*GCKR*) and rs516246 (*FUT2*) are associated with Crohn's disease which might be associated with body composition (S2 Table).

**Table 2. Adjusted associations of liver enzymes (ALT and ALP) with muscle mass, grip strength, and fat percentage at ~17.5 years in the Hong Kong's "Children of 1997" birth cohort, Hong Kong, China.**

| Exposure | Outcome (unit) (N) | Sex-adjusted as confounder | | P-value of interaction with sex | Boys | | Girls | |
|---|---|---|---|---|---|---|---|---|
| | | Beta | 95% CI | | Beta | 95% CI | Beta | 95% CI |
| ALT (IU/L) | Muscle mass (kg) (2520) | 0.11[a] | 0.10 to 0.12 | <0.001 | 0.13[a] | 0.11 to 0.14 | 0.06[a] | 0.04 to 0.07 |
| | Grip strength (kg) (2524) | 0.002 | -0.01 to 0.02 | 0.73 | 0.002 | -0.02 to 0.03 | 0.01 | -0.02 to 0.04 |
| | Fat percentage (2528) | 0.15[a] | 0.13 to 0.17 | 0.19 | 0.16[a] | 0.14 to 0.18 | 0.13[a] | 0.09 to 0.16 |
| ALP (IU/L) | Muscle mass (kg) (2569) | -0.03[a] | -0.04 to -0.02 | <0.001 | -0.04[a] | -0.05 to -0.03 | -0.005 | -0.015 to 0.005 |
| | Grip strength (kg) (2573) | -0.01 | -0.021 to -0.002 | 0.003 | -0.02 | -0.03 to -0.01 | 0.02 | 0.001 to 0.033 |
| | Fat percentage (2577) | -0.02[a] | -0.03 to -0.01 | <0.001 | -0.03[a] | -0.04 to -0.02 | 0.03 | 0.004 to 0.048 |

Adjustment: adjusted for household income, highest parental education, type of housing, highest parental occupation, second-hand and maternal smoking, height and sex.

ALT: alanine aminotransferase; ALP: alkaline phosphatase

[a] Statistically significant in Bonferroni corrected p-value (<0.003) accounting for multiple testing.

**Mendelian randomization estimates.** Table 3 shows similar inverse estimates of genetically predicted ALT with fat-free mass and fat mass from all methods and by sex, however, the confidence intervals included the null value. ALT was not clearly associated with grip strength. Nevertheless, using MR-PRESSO ALT was inversely associated with fat-free mass (-0.41, 95% CI -0.64 to -0.19) and fat mass (-0.58, 95% CI -0.85 to -0.30). Table 4 shows genetically predicted ALP was not clearly associated with fat-free mass, fat mass, or grip strength using any method or by sex. Table 5 shows genetically predicted GGT was not clearly associated with fat-free mass, fat mass or grip strength, but after excluding potential pleiotropy the corrected MR-PRESSO estimates suggested a positive association with fat-free mass (0.30, 95% CI 0.01 to 0.60) and fat mass (0.41, 95% CI 0.10 to 0.71), particularly in women. GGT was not clearly associated with grip strength, although the WM estimate gave positive associations in women.

## Discussion

Using two different complementary designs with different strengths and weaknesses, we examined the impact of liver enzymes on body composition. Although there were discrepancies between the observational and MR estimates, some associations of ALT and GGT with body composition were found.

These two study designs have contrasting limitations. In both the observational and MR designs, we assumed linear associations. Non-linearity cannot be excluded. However, observationally, most participants had clinically normal liver enzymes with a right-skewed distribution. Additionally, using a linear model in MR may be valuable even if the underlying exposure-outcome association is non-linear.[31] Individual data is needed to test for non-linear associations. Observational studies are open to residual confounding, possibly by diet, medication usage, lifestyle, and physical activity, although medication use is rare at 17.5 years and specifically in Hong Kong adolescents, while smoking is rare and alcohol consumption is low. [32–34] Disentangling correlated factors is also difficult in an observational studies. Inevitably, follow-up was incomplete (51%), but participants with and without body composition indices were similar, making selection bias unlikely. We also identified some sex differences which are less open to confounding. Inaccessibility, cost, and exposure to low-dose radiation precluded the use of dual-energy X-ray absorptiometry The reliability of bioimpedance analysis measurements particularly of body fat may vary for many reasons [35] but unlikely with liver function, so any biases are likely towards the null. The discrepancy between the observational and MR

**Table 3. Estimates of the effect of genetically instrumented ALT (per 100% change in concentration) on fat-free mass, fat mass, and grip strength (left and right) using Mendelian randomization with different methodological approaches with and without potentially pleiotropic SNPs and potentially confounded SNPs.**

| Outcome | Sex | SNP[a] | IVW | | | | WM | | MR-Egger | | | MR-PRESSO outlier corrected | | |
|---|---|---|---|---|---|---|---|---|---|---|---|---|---|---|
| | | | Beta | 95% CI | I2 (p-value) | Sex interaction p-value | Beta | 95% CI | Beta | 95% CI | Intercept p-value | Beta | 95% CI | Sex interaction p-value |
| Fat-free Mass (kg) | All | 4 | -0.80 | -2.41 to 0.81 | 91.4% (<0.001) | 0.68 | -0.45 | -0.98 to 0.08 | 0.73 | -2.61 to 4.07 | 0.31 | -0.41 | -0.64 to -0.19 | 0.46 |
| | | 3 | -0.42 | -0.91 to 0.08 | 0.0% (0.81) | 0.48 | -0.43 | -0.96 to 0.09 | -0.75 | -1.91 to 0.41 | 0.53 | - | - | - |
| | | 2 | -0.19 | -1.08 to 0.71 | - | 0.35 | - | - | - | - | - | - | - | - |
| | Male | 4 | -1.10 | -3.20 to 1.00 | 84.1% (<0.001) | - | -0.74 | -1.69 to 0.21 | 0.30 | -4.61 to 5.20 | 0.53 | -0.62 | -1.49 to 0.24 | - |
| | | 3 | -0.62 | -1.49 to 0.25 | 0.0% (0.37) | - | -0.66 | -1.59 to 0.28 | -1.86 | -3.92 to 0.19 | 0.19 | - | - | - |
| | | 2 | 0.30 | -1.29 to 1.88 | - | - | - | - | - | - | - | - | - | - |
| | Female | 4 | -0.55 | -1.87 to 0.76 | 85.3% (<0.001) | - | -0.13 | -0.70 to 0.45 | 1.08 | -1.10 to 3.25 | 0.10 | -0.25 | -0.70 to 0.20 | - |
| | | 3 | -0.25 | -0.77 to 0.27 | 0.0% (0.48) | - | -0.12 | -0.69 to 0.45 | 0.16 | -1.07 to 1.40 | 0.47 | - | - | - |
| | | 2 | -0.59 | -1.55 to 0.37 | - | - | - | - | - | - | - | - | - | - |
| Fat Mass (kg) | All | 4 | -1.22 | -3.89 to 1.46 | 93.2% (<0.001) | 0.90 | -0.59 | -1.38 to 0.19 | 1.41 | -4.02 to 6.84 | 0.28 | -0.58 | -0.85 to -0.30 | 0.57 |
| | | 3 | -0.58 | -1.30 to 0.15 | 0.0% (0.87) | 0.60 | -0.56 | -1.33 to 0.22 | -1.00 | -2.71 to 0.71 | 0.60 | - | - | - |
| | | 2 | -0.28 | -1.60 to 1.04 | - | 0.08 | - | - | - | - | - | - | - | — |
| | Male | 4 | -1.06 | -4.14 to 2.01 | 91.1% (<0.001) | - | -0.49 | -1.52 to 0.53 | 0.95 | -6.24 to 8.13 | 0.53 | -0.36 | -1.58 to 0.85 | - |
| | | 3 | -0.36 | -1.31 to 0.59 | 38.8% (0.20) | - | -0.35 | -1.38 to 0.67 | -2.24 | -4.49 to 0.01 | 0.07 | - | - | - |
| | | 2 | 0.95 | -0.78 to 2.68 | - | - | - | - | - | - | - | - | - | - |
| | Female | 4 | -1.34 | -3.82 to 1.14 | 82.7% (<0.001) | - | -0.79 | -1.94 to 0.37 | 1.82 | -2.08 to 5.72 | 0.07 | -0.76 | -1.32 to -0.20 | - |
| | | 3 | -0.76 | -1.83 to 0.31 | 0.0% (0.76) | - | -0.73 | -1.88 to 0.42 | 0.09 | -2.43 to 2.62 | 0.47 | - | - | - |
| | | 2 | -1.35 | -3.30 to 0.60 | - | - | - | - | - | - | - | - | - | - |
| Left Hand Grip Strength (kg) | All | 4 | 0.00 | -0.57 to 0.57 | 0.0% (0.42) | 0.30 | -0.09 | -0.74 to 0.56 | 0.09 | -1.31 to 1.49 | 0.89 | 0.00[b] | -0.55 to 0.55 | 0.21 |
| | | 3 | 0.08 | -0.52 to 0.67 | 0.0% (0.39) | 0.21 | -0.06 | -0.71 to 0.58 | -0.34 | -2.03 to 1.36 | 0.60 | - | - | - |
| | | 2 | 0.43 | -0.64 to 1.51 | - | 0.38 | - | - | - | - | - | - | - | - |
| | Male | 4 | 0.33 | -0.65 to 1.32 | 0.0% (0.47) | - | 0.27 | -0.84 to 1.38 | 0.71 | -1.51 to 2.94 | 0.70 | 0.33[b] | -0.57 to 1.23 | - |
| | | 3 | 0.49 | -0.53 to 1.51 | 0.0% (0.58) | - | 0.30 | -0.80 to 1.40 | -0.09 | -2.50 to 2.33 | 0.60 | - | - | - |
| | | 2 | 0.99 | -0.87 to 2.85 | - | - | - | - | - | - | - | - | - | - |
| | Female | 4 | -0.30 | -0.93 to 0.34 | 0.0% (0.79) | - | -0.41 | -1.12 to 0.31 | -0.47 | -1.79 to 0.86 | 0.78 | -0.30[b] | -0.67 to 0.08 | - |
| | | 3 | -0.29 | -0.95 to 0.36 | 0.0% (0.59) | - | -0.40 | -1.11 to 0.31 | -0.60 | -2.15 to 0.95 | 0.67 | - | - | - |
| | | 2 | -0.02 | -1.22 to 1.18 | - | - | - | - | - | - | - | - | - | - |

(Continued)

**Table 3.** (Continued)

| Outcome | Sex | SNP[a] | IVW | | | | WM | | MR-Egger | | | MR-PRESSO outlier corrected | | |
|---|---|---|---|---|---|---|---|---|---|---|---|---|---|---|
| | | | Beta | 95% CI | I2 (p-value) | Sex interaction p-value | Beta | 95% CI | Beta | 95% CI | Intercept p-value | Beta | 95% CI | Sex interaction p-value |
| Right Hand Grip Strength (kg) | All | 4 | -0.03 | -0.60 to 0.54 | 0.0% (0.64) | 0.35 | 0.06 | -0.57 to 0.70 | 0.48 | -0.71 to 1.66 | 0.34 | -0.03[b] | -0.46 to 0.40 | 0.25 |
| | | 3 | 0.02 | -0.57 to 0.61 | 0.0% (0.53) | 0.20 | 0.14 | -0.51 to 0.78 | 0.47 | -0.93 to 1.86 | 0.49 | - | - | - |
| | | 2 | -0.25 | -1.32 to 0.83 | - | 0.54 | - | - | - | - | - | - | - | - |
| | Male | 4 | 0.26 | -0.72 to 1.24 | 0.0% (0.48) | - | 0.49 | -0.61 to 1.58 | 1.39 | -0.66 to 3.44 | 0.22 | 0.26[b] | -0.63 to 1.15 | - |
| | | 3 | 0.44 | -0.58 to 1.46 | 0.0% (0.71) | - | 0.57 | -0.53 to1.67 | 0.96 | -1.45 to 3.36 | 0.64 | - | - | - |
| | | 2 | 0.14 | -1.71 to 1.99 | - | - | - | - | - | - | - | - | - | - |
| | Female | 4 | -0.31 | -0.95 to 0.33 | 0.0% (0.79) | - | -0.26 | -0.97 to 0.45 | -0.33 | -1.67 to 1.00 | 0.96 | -0.31[b] | -0.69 to 0.07 | - |
| | | 3 | -0.36 | -1.02 to 0.30 | 0.0% (0.72) | - | -0.27 | -0.98 to 0.45 | 0.00 | -1.56 to 1.56 | 0.62 | - | - | - |
| | | 2 | -0.58 | -1.78 to 0.63 | - | - | - | - | - | - | - | - | - | - |

Potentially pleiotropic and confounded SNP: rs2954021 (*TRIB1*) and rs738409 (*PNPLA3*)

[a] SNP = 4: all SNPs; SNP = 3, excluding rs2954021; SNP = 2, excluding rs738409 additionally

[b] No outlier is found, presenting the raw estimate instead

ALT: alanine aminotransferase; IVW: inverse variance weighting; WM: weighted median; MR-PRESSO: Mendelian Randomization Pleiotropy RESidual Sum and Outlier.

estimates might be due to the difficulty of distinguishing between cause and effect observationally. We additionally obtained an MR estimate of BMI with liver enzymes (ALT, ALP, and GGT), which gave a positive associations of BMI with ALT and ALP but not with GGT, which is consistent with the previously observed positive associations of BMI with liver enzymes.[36–38] (S4 Table) Differences by race/ethnicity are also possible. Lack of relevant data in Chinese precludes examining this possibility. However, we would normally expect causal factors to act consistently unless we know of reasons why the relevance of the specific operating mechanism varies by race/ethnicity.[39] The age difference between the participants in the purely observational and MR designs might also contribute, as body composition may be affected by physical activity, muscle mass, and SEP. However, more parsimoniously it is likely that the drivers of the outcomes are similar, assuming the underlying etiological paths are consistent across age and population. MR assumes the genetic instruments strongly predict the exposure, are not confounded, and are only linked with the outcome by affecting the exposure. The *F* statistics were all >10 suggesting weak instrument bias is unlikely. We repeated the analyses excluding SNPs potentially associated with confounders. Pleiotropic effects are possible, but estimates remained similar after excluding potentially pleiotropic SNPs such as rs738409 (*PNPLA3*) predicting ALT. Additionally, we conducted several sensitivity analyses to assess potential pleiotropy statistically, such as MR-Egger and MR-PRESSO but found no evidence of directional pleiotropy. Given only 4 SNPs predicted ALT, excluding potentially pleiotropic and confounded SNPs would reduce statistical power. The MR estimates were relatively small, which might not be clinically significant, but could be relevant at the population level and may provide etiological insights.[40] The MR analyses were mainly restricted to people of European ancestry. Given the distribution of body composition varies by ethnicity, it is possible that the

**Table 4. Estimates of the effect of genetically instrumented ALP (per 100% change in concentration) on fat-free mass, fat mass, and grip strength (left and right) using Mendelian randomization with different methodological approaches with and without potentially pleiotropic SNPs and potentially confounded SNPs.**

| Outcome | Sex | SNP[a] | IVW | | | | WM | | MR-Egger | | | MR-PRESSO outlier corrected | | |
|---|---|---|---|---|---|---|---|---|---|---|---|---|---|---|
| | | | Beta | 95% CI | I2 (p-value) | Sex interaction p-value | Beta | 95% CI | Beta | 95% CI | Intercept p-value | Beta | 95% CI | Sex interaction p-value |
| Fat-free Mass (kg) | All | 14 | 0.16 | -0.66 to 0.97 | 87.9% (<0.001) | 0.72 | 0.48 | 0.10 to 0.85 | 0.98 | -0.44 to 2.40 | 0.17 | 0.33 | -0.17 to 0.83 | 0.22 |
| | | 11 | -0.002 | -0.98 to 0.97 | 81.9% (<0.001) | 0.86 | 0.28 | -0.34 to 0.90 | 1.26 | -2.06 to 4.59 | 0.43 | 0.19 | -0.45 to 0.83 | 0.98 |
| | | 7 | 0.12 | -1.20 to 1.43 | 86.5% (<0.001) | 0.83 | 0.34 | -0.33 to 1.01 | 1.21 | -3.62 to 6.05 | 0.64 | 0.42 | -0.04 to 0.87 | 0.54 |
| | Male | 14 | 0.29 | -0.73 to 1.32 | 75.9% (<0.001) | - | 0.80 | 0.12 to 1.47 | 1.65 | -0.02 to 3.33 | 0.06 | 0.62 | -0.03 to 1.27 | - |
| | | 11 | -0.09 | -1.28 to 1.11 | 62.0% (<0.003) | - | 0.17 | -0.85 to 1.19 | 0.81 | -3.35 to 4.96 | 0.66 | -0.09 | -0.93 to 0.75 | - |
| | | 7 | -0.01 | -1.29 to 1.28 | 55.7% (0.04) | - | 0.12 | -1.00 to 1.24 | 0.43 | -4.38 to 5.24 | 0.85 | -0.01[b] | -1.29 to 1.28 | - |
| | Female | 14 | 0.04 | -0.78 to 0.86 | 86.3% (<0.001) | - | 0.03 | -0.38 to 0.43 | 0.40 | -1.11 to 1.92 | 0.58 | 0.12 | -0.35 to 0.58 | - |
| | | 11 | 0.07 | -1.01 to 1.14 | 83.1% (<0.001) | - | -0.10 | -0.84 to 0.64 | 1.63 | -2.01 to 5.27 | 0.38 | 0.23 | -0.51 to 0.97 | - |
| | | 7 | 0.22 | -1.29 to 1.73 | 88.4% (<0.001) | - | 0.49 | -0.28 to 1.26 | 1.86 | -3.62 to 7.35 | 0.54 | 0.50 | -0.44 to 1.44 | - |
| Fat Mass (kg) | All | 14 | -0.62 | -1.84 to 0.60 | 88.3% (<0.001) | 0.77 | -0.51 | -1.08 to 0.05 | 0.38 | -1.81 to 2.56 | 0.28 | -0.53 | -0.94 to -0.11 | 0.59 |
| | | 11 | -0.45 | -1.95 to 1.05 | 83.3% (<0.001) | 0.84 | -0.48 | -1.42 to 0.47 | 2.54 | -2.32 to 7.41 | 0.21 | -0.25 | -0.96 to 0.46 | 0.86 |
| | | 7 | -0.35 | -2.54 to 1.84 | 89.4% (<0.001) | 0.94 | -0.27 | -1.37 to 0.84 | 4.61 | -2.26 to 11.48 | 0.14 | -0.05 | -1.00 to 0.90 | 0.68 |
| | Male | 14 | -0.46 | -1.47 to 0.56 | 70.8% (<0.001) | - | -0.23 | -0.95 to 0.48 | 0.54 | -1.24 to 2.31 | 0.19 | -0.24 | -0.78 to 0.31 | - |
| | | 11 | -0.30 | -1.14 to 0.55 | 10.0% (0.35) | - | -0.15 | -1.21 to 0.92 | 1.25 | -1.54 to 4.04 | 0.26 | -0.30[b] | -1.14 to 0.55 | - |
| | | 7 | -0.43 | -1.53 to 0.67 | 27.4% (0.22) | - | -0.62 | -1.83 to 0.59 | 3.46 | 0.25 to 6.66 | 0.01 | -0.43[b] | -1.53 to 0.67 | - |
| | Female | 14 | -0.76 | -2.38 to 0.85 | 85.3% (<0.001) | - | -0.73 | -1.57 to 0.10 | 0.23 | -2.71 to 3.18 | 0.43 | -0.51 | -1.33 to 0.30 | - |
| | | 11 | -0.58 | -2.92 to 1.76 | 85.0% (<0.001) | - | -0.32 | -1.76 to 1.12 | 3.67 | -4.05 to 11.39 | 0.26 | -0.44 | -1.68 to 0.80 | - |
| | | 7 | -0.28 | -3.64 to 3.08 | 90.2% (<0.001) | - | 0.56 | -1.00 to 2.12 | 5.63 | -5.78 to 17.03 | 0.29 | -0.02 | -1.54 to 1.49 | - |
| Left Hand Grip Strength (kg) | All | 14 | 0.61 | 0.04 to 1.18 | 64.6% (<0.001) | 0.66 | 0.93 | 0.41 to 1.44 | 1.48 | 0.59 to 2.37 | 0.02 | 0.76 | 0.29 to 1.22 | 0.59 |
| | | 11 | 0.10 | -0.66 to 0.85 | 56.1% (0.01) | 0.35 | 0.08 | -0.64 to 0.80 | 0.20 | -2.45 to 2.85 | 0.94 | 0.38 | -0.22 to 1.00 | 0.27 |
| | | 7 | 0.13 | -0.45 to 0.71 | 0.0% (0.59) | 0.86 | 0.31 | -0.45 to 1.07 | 0.91 | -1.08 to 2.90 | 0.42 | 0.13[b] | -0.37 to 0.64 | 0.45 |
| | Male | 14 | 0.49 | -0.31 to 1.28 | 44.9% (0.04) | - | 1.30 | 0.46 to 2.13 | 2.11 | 1.05 to 3.17 | 0.00 | 0.69 | 0.03 to 1.35 | - |
| | | 11 | -0.26 | -1.19 to 0.68 | 14.7% (0.30) | - | 0.28 | -0.91 to 1.47 | 1.59 | -1.45 to 4.63 | 0.21 | -0.26[b] | -1.19 to 0.68 | - |
| | | 7 | 0.06 | -0.94 to 1.07 | 0.0% (0.77) | - | 0.32 | -0.92 to 1.57 | 1.62 | -1.82 to 5.07 | 0.35 | 0.06[b] | -0.68 to 0.81 | - |
| | Female | 14 | 0.72 | 0.09 to 1.36 | 64.2% (<0.001) | - | 1.02 | 0.49 to 1.55 | 0.95 | -0.23 to 2.13 | 0.65 | 0.75 | 0.26 to 1.24 | - |
| | | 11 | 0.41 | -0.60 to 1.42 | 69.7% (<0.001) | - | 0.56 | -0.35 to 1.47 | -1.03 | -4.45 to 2.39 | 0.39 | 0.44 | -0.36 to 1.23 | - |
| | | 7 | 0.20 | -0.84 to 1.24 | 61.2% (0.02) | - | 0.51 | -0.50 to 1.51 | 0.25 | -3.66 to 4.15 | 0.98 | 0.51 | -0.37 to 1.39 | - |

*(Continued)*

**Table 4.** (Continued)

| Outcome | Sex | SNP[a] | IVW | | | | WM | | MR-Egger | | | MR-PRESSO outlier corrected | | |
|---|---|---|---|---|---|---|---|---|---|---|---|---|---|---|
| | | | Beta | 95% CI | I2 (p-value) | Sex interaction p-value | Beta | 95% CI | Beta | 95% CI | Intercept p-value | Beta | 95% CI | Sex interaction p-value |
| Right Hand Grip Strength (kg) | All | 14 | 0.46 | -0.22 to 1.15 | 75.4% (<0.001) | 0.55 | 1.32 | 0.74 to 1.90 | 1.58 | 0.54 to 2.62 | 0.01 | -0.12 | -0.81 to 0.58 | 0.001 |
| | | 11 | -0.23 | -1.06 to 0.60 | 63.9% (0.002) | 0.09 | -0.39 | -1.18 to 0.40 | -0.28 | -3.21 to 2.64 | 0.97 | -0.12 | -0.76 to 0.51 | 0.04 |
| | | 7 | -0.22 | -0.94 to 0.50 | 35.1% (0.16) | 0.11 | -0.50 | -1.33 to 0.34 | 0.63 | -1.96 to 3.22 | 0.50 | -0.22[b] | -0.94 to 0.50 | 0.11 |
| | Male | 14 | 0.27 | -0.69 to 1.24 | 62.9% (<0.001) | - | 0.56 | -0.47 to 1.59 | 2.26 | 1.08 to 3.45 | 0.00 | -1.02 | -1.94 to -0.09 | - |
| | | 11 | -0.86 | -1.88 to 0.16 | 28.9% (0.17) | - | -0.81 | -2.01 to 0.40 | 0.44 | -3.04 to 3.92 | 0.44 | -0.86[b] | -1.88 to 0.16 | - |
| | | 7 | -0.82 | -1.91 to 0.28 | 16.3% (0.31) | - | -0.71 | -2.00 to 0.59 | 1.04 | -2.70 to 4.78 | 0.31 | -0.81[b] | -1.91 to 0.28 | - |
| | Female | 14 | 0.64 | 0.01 to 1.26 | 62.2% (0.001) | - | 0.97 | 0.44 to 1.51 | 1.00 | -0.14 to 2.13 | 0.45 | 0.70 | 0.22 to 1.18 | - |
| | | 11 | 0.32 | -0.62 to 1.27 | 64.7% (0.002) | - | 0.72 | -0.18 to 1.63 | -0.94 | -4.14 to 2.26 | 0.42 | 0.43 | -0.28 to 1.14 | - |
| | | 7 | 0.30 | -0.52 to 1.12 | 37.2% (0.14) | - | 0.67 | -0.29 to 1.61 | 0.22 | -2.87 to 3.31 | 0.96 | 0.30[b] | -0.52 to 1.12 | - |

Potentially pleiotropic SNPs: rs281377 (*FUT2*), rs2954021 (*TRIB1*), and rs579459 (*ABO*)

Potentially confounded SNPs: rs174601 (*C11orf10, FADS1, FADS2*), rs2236653 (*ST3GAL4*) rs281377 (*FUT2*), rs2954021 (*TRIB1*), rs579459 (*ABO*), rs6984305 (*PPP1R3B*), and rs7923609 (*JMJD1C, NRBF2*)

[a] SNP = 14: all SNPs; SNP = 11, excluding potentially pleiotropic SNPs; SNP = 7, excluding potentially pleiotropic SNPs and potentially confounded SNPs

[b] No outlier is found, presenting the raw estimate instead

ALP: alkaline phosphatase; IVW: inverse variance weighting; WM: weighted median; MR-PRESSO: Mendelian Randomization Pleiotropy RESidual Sum and Outlier

drivers of body composition also vary by ethnicity. However, more parsimoniously, it is likely that the drivers of body composition are similar across populations but their relevance varies. Specifically, ALT is higher in Chinese than in Westerners [41] which might be relevant to the lower fat-free mass in Chinese than in Westerners, [42] although ethnic variation in both ALT and fat-free mass could just be due to chance. The use of summary statistics in the MR study means we could not comprehensively assess the differences by age, sex or by baseline levels of liver enzymes; but we assessed the differences by sex observationally. Replicating the MR study in a Chinese population would be very helpful. However, publicly available large GWAS of liver enzymes or body composition in Chinese does not exist. Liver enzymes might not completely or only represent liver function, for example, ALT may be transitorily affected by physical exertion, but liver enzymes are widely used as a surrogate of liver function.[20] Here, SNPs associated with vigorous physical activity were excluded. Fat-free mass and muscle mass are not identical. Fat-free mass also includes organs, skin, bones and body water, but does not vary as much as muscle mass. Finally, some overlap of participants between the GWAS used is inevitable, however, any effect on the estimates is likely to be small.

These observations are similar to previous observational studies.[10, 43, 44] However, only some of the previous observations, i.e., higher ALT associated with lower fat-free mass [45] and higher GGT associated with adiposity [46, 47] were confirmed using MR. Being consistent with observational studies, some differences by sex were found.[43, 44, 47]

Compared with a purely observational design, MR studies are more suitable for assessing the direction of causality, given genetic variants are randomly allocated at conception.[2, 48] Etiologically, the association of higher ALT, a measure of hepatocyte integrity, with lower fat-

**Table 5. Estimates of the effect of genetically instrumented GGT (per 100% change in concentration) on fat-free mass, fat mass, and grip strength (left and right) using Mendelian randomization with different methodological approaches with and without potentially pleiotropic SNPs and potentially confounded SNPs.**

| Outcome | Sex | SNP[a] | IVW | | | | WM | | MR-Egger | | | MR-PRESSO outlier corrected | | |
|---|---|---|---|---|---|---|---|---|---|---|---|---|---|---|
| | | | Beta | 95% CI | I2 (p-value) | Sex interaction p-value | Beta | 95% CI | Beta | 95% CI | Intercept p-value | Beta | 95% CI | Sex interaction p-value |
| Fat-free Mass (kg) | All | 26 | -0.02 | -0.63 to 0.58 | 94.7% (<0.001) | 0.79 | 0.07 | -0.16 to 0.30 | 0.35 | -1.01 to 1.72 | 0.55 | 0.17 | -0.08 to 0.42 | 0.30 |
| | | 23 | 0.16 | -0.34 to 0.66 | 91.7% (<0.001) | 0.91 | 0.10 | -0.13 to 0.33 | -0.01 | -1.14 to 1.12 | 0.75 | 0.17 | -0.10 to 0.43 | 0.33 |
| | | 17 | 0.33 | -0.05 to 0.71 | 81.4% (<0.001) | 0.97 | 0.18 | -0.08 to 0.45 | 0.13 | -0.67 to 0.93 | 0.59 | 0.30 | 0.01 to 0.60 | 0.64 |
| | Male | 26 | -0.09 | -0.90 to 0.71 | 90.7% (<0.001) | - | -0.01 | -0.42 to 0.40 | 0.24 | -1.59 to 2.07 | 0.69 | -0.03 | -0.46 to 0.40 | - |
| | | 23 | 0.13 | -0.51 to 0.78 | 84.5% (<0.001) | - | 0.01 | -0.39 to 0.42 | -0.19 | -1.65 to 1.27 | 0.63 | 0.07 | -0.35 to 0.49 | - |
| | | 17 | 0.34 | -0.25 to 0.93 | 75.9% (<0.001) | - | 0.20 | -0.25 to 0.64 | 0.01 | -1.24 to 1.25 | 0.55 | 0.25 | -0.16 to 0.67 | - |
| | Female | 26 | 0.04 | -0.43 to 0.52 | 90.2% (<0.001) | - | 0.14 | -0.10 to 0.38 | 0.45 | -0.61 to 1.52 | 0.39 | 0.21 | -0.07 to 0.48 | - |
| | | 23 | 0.18 | -0.23 to 0.60 | 86.6% (<0.001) | - | 0.15 | -0.09 to 0.38 | 0.16 | -0.79 to 1.11 | 0.95 | 0.32 | 0.06 to 0.58 | - |
| | | 17 | 0.32 | 0.05 to 0.59 | 58.2% (0.001) | - | 0.21 | -0.05 to 0.46 | 0.24 | -0.33 to 0.82 | 0.76 | 0.36 | 0.11 to 0.61 | - |
| Fat Mass (kg) | All | 26 | 0.22 | -0.24 to 0.67 | 79.5% (<0.001) | 0.30 | 0.22 | -0.12 to 0.55 | 0.39 | -0.64 to 1.41 | 0.71 | 0.11 | -0.24 to 0.46 | 0.35 |
| | | 23 | 0.27 | -0.18 to 0.72 | 78.3% (<0.001) | 0.30 | 0.22 | -0.11 to 0.55 | 0.19 | -0.84 to 1.23 | 0.87 | 0.04 | -0.28 to 0.37 | 0.16 |
| | | 17 | 0.45 | -0.04 to 0.94 | 75.4% (<0.001) | 0.49 | 0.24 | -0.11 to 0.58 | 0.27 | -0.76 to 1.30 | 0.70 | 0.41 | 0.10 to 0.71 | 0.17 |
| | Male | 26 | -0.01 | -0.52 to 0.51 | 73.0% (<0.001) | - | 0.06 | -0.37 to 0.49 | 0.14 | -1.03 to 1.32 | 0.78 | -0.03 | -0.39 to 0.33 | - |
| | | 23 | 0.06 | -0.37 to 0.48 | 58.2% (<0.001) | - | 0.06 | -0.36 to 0.49 | -0.11 | -1.08 to 0.87 | 0.72 | -0.01 | -0.31 to 0.28 | - |
| | | 17 | 0.30 | -0.20 to 0.81 | 60.6% (<0.001) | - | 0.09 | -0.36 to 0.53 | 0.02 | -1.04 to 1.09 | 0.56 | 0.23 | -0.06 to 0.52 | - |
| | Female | 26 | 0.40 | -0.17 to 0.97 | 72.0% (<0.001) | - | 0.36 | -0.12 to 0.83 | 0.60 | -0.69 to 1.90 | 0.73 | 0.36 | -0.13 to 0.84 | - |
| | | 23 | 0.45 | -0.16 to 1.07 | 74.1% (<0.001) | - | 0.37 | -0.10 to 0.84 | 0.45 | -0.94 to 0.19 | 1.00 | 0.41 | -0.11 to 0.93 | - |
| | | 17 | 0.58 | 0.004 to 1.147 | 61.0% (<0.001) | - | 0.41 | -0.09 to 0.92 | 0.49 | -0.73 to 1.70 | 0.87 | 0.65 | 0.12 to 1.18 | - |
| Left Hand Grip Strength (kg) | All | 26 | 0.06 | -0.26 to 0.38 | 73.2% (<0.001) | 0.71 | 0.17 | -0.08 to 0.43 | 0.26 | -0.47 to 0.98 | 0.56 | 0.19 | -0.01 to 0.39 | 0.47 |
| | | 23 | 0.09 | -0.23 to 0.41 | 70.9% (<0.001) | 0.81 | 0.18 | -0.07 to 0.43 | 0.17 | -0.55 to 0.90 | 0.81 | 0.17 | -0.04 to 0.38 | 0.56 |
| | | 17 | 0.22 | -0.01 to 0.45 | 25.4% (0.16) | 0.49 | 0.23 | -0.05 to 0.51 | 0.25 | -0.23 to 0.74 | 0.88 | 0.22[b] | -0.01 to 0.45 | 0.31 |
| | Male | 26 | 0.01 | -0.42 to 0.43 | 53.3% (<0.001) | - | -0.01 | -0.45 to 0.42 | 0.11 | -0.85 to 1.07 | 0.82 | 0.15 | -0.20 to 0.50 | - |
| | | 23 | 0.06 | -0.35 to 0.47 | 47.5% (0.007) | - | -0.02 | -0.45 to 0.41 | 0.03 | -0.90 to 0.97 | 0.96 | 0.07 | -0.26 to 0.40 | - |
| | | 17 | 0.13 | -0.23 to 0.49 | 12.1% (0.31) | - | 0.04 | -0.43 to 0.50 | 0.19 | -0.58 to 1.00 | 0.86 | 0.13[b] | -0.23 to 0.49 | - |
| | Female | 26 | 0.12 | -0.23 to 0.47 | 71.8% (<0.001) | - | 0.47 | 0.18 to 0.76 | 0.39 | -0.40 to 1.18 | 0.44 | 0.25 | -0.03 to 0.52 | - |
| | | 23 | 0.13 | -0.24 to 0.49 | 72.4% (<0.001) | - | 0.47 | 0.18 to 0.77 | 0.31 | -0.52 to 1.13 | 0.64 | 0.27 | -0.0002 to 0.54 | - |
| | | 17 | 0.30 | -0.01 to 0.62 | 51.8% (0.007) | - | 0.50 | 0.17 to 0.82 | 0.32 | -0.36 to 1.00 | 0.97 | 0.3[b] | -0.01 to 0.62 | - |

(Continued)

**Table 5.** (Continued)

| Outcome | Sex | SNP[a] | IVW | | | | WM | | MR-Egger | | | MR-PRESSO outlier corrected | | |
|---|---|---|---|---|---|---|---|---|---|---|---|---|---|---|
| | | | Beta | 95% CI | I2 (p-value) | Sex interaction p-value | Beta | 95% CI | Beta | 95% CI | Intercept p-value | Beta | 95% CI | Sex interaction p-value |
| Right Hand Grip Strength (kg) | All | 26 | 0.01 | -0.33 to 0.34 | 75.3% (<0.001) | 0.65 | 0.16 | -0.09 to 0.41 | 0.13 | -0.63 to 0.89 | 0.72 | 0.14 | -0.06 to 0.34 | 0.14 |
| | | 23 | 0.04 | -0.29 to 0.36 | 72.2% (<0.001) | 0.69 | 0.16 | -0.09 to 0.41 | 0.09 | -0.65 to 0.83 | 0.88 | 0.11 | -0.09 to 0.32 | 0.45 |
| | | 17 | 0.16 | -0.04 to 0.35 | 0.0% (0.48) | 0.34 | 0.17 | -0.11 to 0.44 | 0.15 | -0.26 to 0.57 | 1.00 | 0.15[b] | -0.04 to 0.34 | 0.33 |
| | Male | 26 | -0.07 | -0.49 to 0.36 | 54.4% (<0.001) | - | -0.11 | -0.55 to 0.33 | 0.10 | -0.87 to 1.07 | 0.71 | -0.05 | -0.38 to 0.28 | - |
| | | 23 | -0.03 | -0.46 to 0.41 | 53.1% (0.002) | - | -0.11 | -0.54 to 0.32 | 0.09 | -0.90 to 1.07 | 0.80 | 0.11 | -0.09 to 0.32 | - |
| | | 17 | 0.04 | -0.30 to 0.38 | 0.0% (0.55) | - | -0.11 | -0.58 to 0.37 | 0.22 | -0.48 to 0.92 | 0.56 | 0.04[b] | -0.29 to 0.36 | - |
| | Female | 26 | 0.07 | -0.30 to 0.44 | 74.3% (<0.001) | - | 0.42 | 0.13 to 0.71 | 0.16 | -0.68 to 1.00 | 0.81 | 0.27 | 0.01 to 0.52 | - |
| | | 23 | 0.09 | -0.27 to 0.46 | 72.3% (<0.001) | - | 0.40 | 0.11 to 0.69 | 0.10 | -0.74 to 0.93 | 1.00 | 0.24 | -0.03 to 0.51 | - |
| | | 17 | 0.26 | -0.04 to 0.56 | 46.3% (0.02) | - | 0.36 | 0.04 to 0.68 | 0.11 | -0.53 to 0.74 | 0.59 | 0.26[b] | -0.04 to 0.56 | - |

Potentially pleiotropic SNPs: rs12968116 (*ATP8B1*), rs1260326 (*GCKR*), and rs516246 (*FUT2*)

Potentially confounded SNPs: rs10908458 (*DPM3, EFNA1, PKLR*), rs12145922 (*CCBL2, PKN2*), rs1260326 (*GCKR*), rs1497406 (*RSG1, EPHA2*), rs17145750 (*MLXIPL*), rs516246 (*FUT2*), rs7310409 (*HNF1A, C12orf27*), and rs754466 (*DLG5*)

[a] SNP = 26: all SNPs; SNP = 23, excluding potentially pleiotropic SNPs; SNP = 17, excluding potentially pleiotropic SNPs and potentially confounded SNPs

[b] No outlier is found, presenting the raw estimate instead

GGT: gamma glutamyltransferase; IVW: inverse variance weighting; WM: weighted median; MR-PRESSO: Mendelian Randomization Pleiotropy RESidual Sum and Outlier.

free mass, possibly differing by sex, may be due to growth hormone (GH) / insulin-like growth factor 1 (IGF-1) or sex hormones which are associated with chronic liver diseases and muscle mass.[49–52] Studies using IGF-1 gene knock out animal models suggest IGF-1 is associated with hyperinsulinaemia and muscle insulin insensitivity, [53–55] although whether GH/IGF-1 also specifically affects ALT and muscle mass overall or differentially by sex is unknown. Schooling et al. have previously suggested that lower levels of androgens might cause higher risk of diabetes via lower muscle mass [51] and poor liver function may reduce androgens, [52] consistent with the sex differences observed. Additionally, it is also consistent with statins usage which is associated with lower testosterone, [56] elevated aminotransferase levels, [57] and higher diabetes risk.[58] Etiologically, these findings are consistent with the evolutionary public health, i.e., growth and reproduction trading-off against longevity, which may inform the identification of interventions. Reasons for an inverse association of ALT with fat mass are unclear since fat mass is a well-established causal factor for diabetes which is unlikely to be contributing to the positive association of ALT with diabetes seen here.[5, 6] But these inverse estimates are consistent with a previous MR study [5] showing ALT negatively associated with BMI using the same genetic variants predicting ALT applied to the 2018 GIANT and UK Bio-bank meta-analysis. The positive associations of GGT with body composition, in particular with fat mass, might be relevant to the observed associations of GGT with cardiovascular disease risk.[59–61] However, no causal role of GGT in cardiovascular disease was found in an MR study.[3]

## Conclusion

Higher ALT, representing hepatocyte integrity, could reduce fat-free mass and fat mass with differences by sex; whilst higher GGT, as a marker of cholestasis, might increase fat-free mass and fat mass. As such, our study provides some indications that lower fat-free mass may mediate the positive effect of ALT on diabetes risk, which requires confirmation in other studies.

## Supporting information

**S1 Table. Baseline characteristics of the participants who were included (n = 3,455) and excluded (n = 4,872) in the analyses of the Hong Kong's "Children of 1997" birth cohort, Hong Kong, China, 1997 to 2016.**
(DOCX)

**S2 Table. Single nucleotide polymorphisms (SNPs) with potential pleiotropic effects other than via the specific liver enzyme from Ensembl, GWAS Catalog, and potential confounders from UK Biobank.**
(DOCX)

**S3 Table. Characteristics of palindromic single nucleotide polymorphisms (SNPs) in the exposure and outcome genome-wide association studies.**
(DOCX)

**S4 Table. Estimates of the effect of genetically instrumented BMI on ALT, ALP, and GGT using Mendelian randomization with different methodological approaches.**
(DOCX)

## Acknowledgments

The authors thank colleagues at the Student Health Service and Family Health Service of the Department of Health for their assistance and collaboration. They also thank late Dr. Connie O for coordinating the project and all the fieldwork for the initial study in 1997–1998.

## Author Contributions

**Conceptualization:** Shiu Lun Au Yeung, C. Mary Schooling.

**Data curation:** Man Ki Kwok, June Yue Yan Leung, Lai Ling Hui.

**Formal analysis:** Junxi Liu.

**Investigation:** Junxi Liu.

**Supervision:** Shiu Lun Au Yeung, Gabriel Matthew Leung, C. Mary Schooling.

**Writing – original draft:** Junxi Liu.

**Writing – review & editing:** Shiu Lun Au Yeung, C. Mary Schooling.

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
