## [Decision Letter · Decision Letter 0]

5 Dec 2019

PONE-D-19-28757

The effect of liver enzymes on body composition: a Mendelian randomization study

PLOS ONE

Dear Dr Schooling,

Thank you for submitting your manuscript to PLOS ONE. After careful consideration, we feel that it has merit but does not fully meet PLOS ONE’s publication criteria as it currently stands. Therefore, we invite you to submit a revised version of the manuscript that addresses the points raised during the review process.

We would appreciate receiving your revised manuscript by Jan 19 2020 11:59PM. To enhance the reproducibility of your results, we recommend that if applicable you deposit your laboratory protocols in protocols.io, where a protocol can be assigned its own identifier (DOI) such that it can be cited independently in the future. For instructions see: http://journals.plos.org/plosone/s/submission-guidelines#loc-laboratory-protocols

We look forward to receiving your revised manuscript.

Kind regards,

David Meyre

Academic Editor

PLOS ONE

Journal Requirements:

3. We noted in your submission details that a portion of your manuscript may have been presented or published elsewhere. Please clarify whether this (PMID: 31504067) was peer-reviewed and formally published. If this work was previously peer-reviewed and published, in the cover letter please provide the reason that this work does not constitute dual publication and should be included in the current manuscript.

Additional Editor Comments (if provided):

Reviewers' comments:

Reviewer's Responses to Questions

**Comments to the Author**

1. Is the manuscript technically sound, and do the data support the conclusions?

Reviewer #1: Yes

Reviewer #2: Partly

2. Has the statistical analysis been performed appropriately and rigorously? 

Reviewer #1: Yes

Reviewer #2: I Don't Know

3. Have the authors made all data underlying the findings in their manuscript fully available?

Reviewer #1: No

Reviewer #2: Yes

4. Is the manuscript presented in an intelligible fashion and written in standard English?

Reviewer #1: Yes

Reviewer #2: No

5. Review Comments to the Author

Reviewer #1: The authors studied the association between liver enzymes and body composition in an observational setting and Mendelian Randomization analysis. The authors provided evidence that specifically ALT was associated with a lower fat mass and fat free mass. This is an interesting, well-written and well-conducted study.

I have a few comments that need to be addressed to further improve the paper:

- In both the observational and Mendelian Randomization analyses, the authors assume a linear association between the liver enzymes and the measures of body composition. However, there is increasing evidence this is not the case. For example, meta-analyses of liver enzymes and diabetes mellitus show a clear non-linear association between determinant and outcome. Is there any evidence in the current (specifically observational analyses) that the association was linear or non-linear?

- The authors performed an observational analysis in relatively young individuals, whereas the Mendelian Randomization analysis is done in relatively older individuals from the UK Biobank population. This is difficult to compare. From literature it is known that specifically body composition at younger age has a different biological background than at older age (e.g., adult weight gain, muscle athrophy etc). The paper would benefit from a comparison of more comparable study populations (e.g., include another study of middle-aged individuals?).

- The authors performed Mendelian Randomization analyses stratified by sex. However, the genetic instruments were retrieved not sex-specific. Can the authors command on this? Is there evidence that the genetic background is similar for the sexes?

- The authors should provide more details on the used methodology to measure the liver enzymes in blood samples and how the body composition was measures. Furthermore, more details should be given on the software used for the present project.

- For comparison of the observational and MR findings, it would be good that both analyses are performed in the same unit. The GWAS (and thus the unit of the MR analysis) was log transformed.

- Line 129: "small proportion". Can the authors provide a percentrage?

- The authors might was to think of a different way to present the data. Given the very small effect sizes, results are hard to interpret.

- The conclusion of the manuscript is heavily weighted on the fat-free mass findings, but do not take into account the additional findings with fat mass. The original aim was that the previously observed association between ALT and BMI will be further explored using fat mass and fat-free mass. However, the authors should note that both outcomes show rather similar results.

- The authots should acknowledge the fact the ALT has a limited number of intruments for the MR analyses (specifically after excluding potential pleiotropic SNPs).

Reviewer #2: Liu XJ et al. assessed the associations of liver enzymes with muscle and fat mass using a two-sample MR. Authors find that higher ALT could reduce fat-free mass and fat mass, while higher GGT, as a might increase fat-free mass and fat mass. Also, authors report sex differences in the association of markers of liver functions and body composition.

Major comment:

-Do authors have information on cardiovascular risk factors such as blood pressure, glucose and blood lipids? If yes, then authors should consider adjusting for these factors.

-In the observational study authors should adjust for smoking and alcohol intake. A sensitivity analysis should e performed among non-smokers and non-drinkers.

-Authors select the SNPs based on Caucasians, while the observational association is explored in Chinese population. Liver function and obesity differs by ethnicity; do authors have an explanation for their choice? Did they test the association of selected SNPs with ALT/AST/GGT in Asian population? Is there any published GWAS on markers of liver function in Asian population?

-Was there information on medications that could alter liver functions? These should be considered in the observational analysis.

-Authors should discuss further what could the reason of the sex-differences they find.

A previous MR study reported no causal effect of GGT on prediabetes and diabetes (Nano et al. 2017) in both men and women. In the current study, there is a suggestive causal effect of GGT on fat mass, particularly in women. What would be the implications of such findings?

-In the obersvational study authors do not invesitage the association between GGT and body composition parameters, which limits their MR findings.

Minor comments

-When describing the results, authors should provide estimates and not just reporting there was or not an association.

-The manuscripts needs English Editing. Also in the method section, authors should descrbie first the observational association and then the MR.

6. PLOS authors have the option to publish the peer review history of their article (what does this mean?). If published, this will include your full peer review and any attached files.

Reviewer #1: No

Reviewer #2: No

---

## [Author Response · Author response to Decision Letter 0]

14 Jan 2020

Thank you so much indeed for the helpful comments. 

Details responses are present in the Response to the Reviewers file.

---

## [Editor Report · Decision Letter 1]

23 Jan 2020

The effect of liver enzymes on body composition: a Mendelian randomization study

PONE-D-19-28757R1

Dear Dr. Schooling,

We are pleased to inform you that your manuscript has been judged scientifically suitable for publication and will be formally accepted for publication once it complies with all outstanding technical requirements.

With kind regards,

David Meyre

Academic Editor

PLOS ONE
---

## [Editor Report · Acceptance letter]

3 Feb 2020

PONE-D-19-28757R1 

The effect of liver enzymes on body composition: a Mendelian randomization study 

Dear Dr. Schooling:

I am pleased to inform you that your manuscript has been deemed suitable for publication in PLOS ONE. Congratulations! Your manuscript is now with our production department. 

With kind regards,

on behalf of

Dr David Meyre 

Academic Editor

PLOS ONE